

# One becomes two: second species of the *Euwallacea fornicatus* (Coleoptera: Curculionidae: Scolytinae) species complex is established on two Hawaiian Islands

Paul F. Rugman-Jones[1], Michelle Au[2], Valeh Ebrahimi[1], Akif Eskalen[3], Conrad P.D.T. Gillett[2], David Honsberger[2], Deena Husein[1], Mark G. Wright[2], Fazila Yousuf[2,4] and Richard Stouthamer[1]

[1] Department of Entomology, University of California, Riverside, Riverside, CA, United States of America
[2] Department of Plant & Environmental Protection Sciences, University of Hawai'i at Manoa, Honolulu, Hawai'i, United States of America
[3] Department of Plant Pathology, University of California, Davis, Davis, CA, United States of America
[4] USDA-ARS, Daniel Inouye Pacific Basin Agricultural Research Center, Hilo, Hawai'i, United States of America

Corresponding author
Paul F. Rugman-Jones,
paulrj@ucr.edu

## ABSTRACT

The cryptic species that make up the *Euwallacea fornicatus* species complex can be readily distinguished via their DNA sequences. Until recently, it was believed that the Hawaiian Islands had been invaded by only one of these cryptic species, *E. perbrevis* (tea shot hole borer; TSHB). However, following the 2016 deposition of a DNA sequence in the public repository GenBank, it became evident that another species, *E. fornicatus* (polyphagous shot hole borer; PSHB), had been detected in macadamia orchards on Hawai'i Island (the Big Island). We surveyed the two most-populous islands of Hawai'i, Big Island and O'ahu, and herein confirm that populations of TSHB and PSHB are established on both. Beetles were collected using a variety of techniques in macadamia orchards and natural areas. Individual specimens were identified to species using a high-resolution melt assay, described herein and validated by subsequent sequencing of specimens. It remains unclear how long each species has been present in the state, and while neither is currently recognized as causing serious economic or ecological damage in Hawai'i, the similarity of the newly-confirmed PSHB population to other damaging invasive PSHB populations around the world is discussed. Although the invasive PSHB populations in Hawai'i and California likely have different geographic origins within the beetle's native range, they share identical *Fusarium* and *Graphium* fungal symbionts, neither of which have been isolated from PSHB in that native range.

## INTRODUCTION

Species of the *Euwallacea fornicatus* complex attracted attention following their invasion and establishment in California and Florida. At the time of these invasions, the complex

was thought to be a single species (*Wood & Bright, 1992*), but after their emergence as significant pests in agricultural and natural ecosystems in the respective states, the invasive beetles were shown to be different species, and moreover, *E. fornicatus s. l.* was unveiled as a complex of at least four cryptic species (*Stouthamer et al., 2017*). The earliest records of this species complex in the 48 contiguous states stem from collections made in 2003 (California) and 2004 (Florida) (*Rabaglia, Dole & Cognato, 2006*). However, the island state of Hawaiʻi was invaded much earlier, with collections of *Xyleborus* (= *Euwallacea*) *fornicatus* existing from the early part of the 20th century. The earliest confirmed collections are from avocado on Oʻahu in 1910 (*Swezey, 1941*), but the author also states that it was known from avocado for many years prior to this. The presence of *E. fornicatus* was subsequently confirmed on the Big Island Hawaiʻi (1919), Maui (1930), and Molokaʻi (1936) (*Swezey, 1941*; *Schedl, 1941*). *Samuelson (1981)* later added Kauaʻi to the list but without a date. Thus, it appears that the beetles invaded the state sometime before 1910 and have since spread to all the islands.

In addition to these three U.S. states, invasive populations of beetles morphologically identified as *E. fornicatus* have successfully established in many other places outside of their native range in Asia. They have been reported as invasive in the following locations (*CABI, 2020*): Australia, Papua New Guinea, Vanuatu, Fiji, Solomon Islands, Micronesia, Samoa, Niue, Hawaiʻi, Comoros, Madagascar, Reunion, South Africa, Israel, Costa Rica, Guatemala, Mexico, Panama, and the continental USA. Exactly how CABI determined the invasive status of these beetles is not clear, and both Australia and Papua New Guinea may in fact prove to be in the native range of this species complex (*Stouthamer et al., 2017*). Like other ambrosia beetles, members of the *E. fornicatus* species complex are particularly well-equipped to invade new geographical areas. Female beetles excavate individual tunnels (galleries) inside branches and trunks of trees. Inside the gallery, the female inoculates the walls with symbiotic fungi and lays her eggs. The fungi grow, extracting nutrients from the plant, and these fungi provide the sole food source for the mother and her developing brood. In this state, the beetles can survive long distance transport very well. On reaching adulthood, female offspring leave the natal gallery, taking with them spores of the symbiotic fungi stored inside special organs called mycangia. The invasive potential of ambrosia beetles is further boosted by their sex determination mechanism and mating system. Like bees and ants, these beetles are haplo-diploid; males develop from unfertilized haploid eggs and females from fertilized diploid eggs (*Kirkendall, 1993*). Their mating system is an example of local mate competition, where mothers produce many daughters and only a few sons. Thus, daughters mate with a brother (sib-mating) inside the natal gallery, and upon leaving, are already inseminated prior to dispersal through the environment. In the *E. fornicatus* species complex, dispersal can be through flight or simply by creating new galleries on the trees where they were born (*Calnaido, 1965*). Therefore, in contrast to many other species where colonization of a new environment may be constrained by the need to meet members of the opposite sex, the population growth rate of this complex is not limited by lack of mates.

As already mentioned, the species morphologically recognized as *E. fornicatus* has recently been shown to consist of several cryptic species. Confirmation of these species
was based on the discovery of substantial differences in the DNA sequences of multiple genes among a worldwide sample of populations (*Stouthamer et al., 2017*). Four DNA lineages were recognized that were initially given the common names of tea shot hole borer (TSHB) 1A and 1B, polyphagous shot hole borer (PSHB) and Kuroshio shot hole borer (KSHB). These different species could be easily recognized by the DNA sequence of the mitochondrial cytochrome oxidase 1 (COI) gene. In the same study, *Stouthamer et al. (2017)* found that the populations they sampled from the Big Island and Maui were genetically identical, belonging to the TSHB-1B lineage. They were also identical to invasive populations in Florida, but differed from those in California (identified as PSHB and KSHB). Thus, TSHB was thought to be the only species of the *E. fornicatus* species complex to have invaded Hawai'i (*Stouthamer et al., 2017*). Recent attempts to associate existing junior synonyms with these species (*Gomez et al., 2018*; *Smith et al., 2019*) have resulted in the current association of the scientific name *E. perbrevis* with this species (*Smith et al., 2019*). Following the publication of the *Stouthamer et al. (2017)* study, we discovered that in September 2016, a conflicting COI sequence was belatedly deposited in GenBank, which originated from two beetles collected from macadamia trees on the Big Island. The COI sequence identified these beetles as *E. fornicatus* (*Smith et al., 2019*) (or PSHB as we choose to refer to it), and they were collected in 2007 by Australian scientists studying the pests attacking macadamia trees in Hawai'i (*Mitchell & Maddox, 2010*). For simplicity, hereafter we refer to *E. perbrevis* and *E. fornicatus* as TSHB and PSHB, respectively. We determine if PSHB is established on the Big Island, and also if it is present on Oʻahu, and identify fungal species associated with these beetles in Hawai'i.

## MATERIAL & METHODS

### Specimen collection

Specimens were collected in natural areas under permissions granted to CG by the United States Department of the Interior National Park Service (permit # HAVO-2019-SCI-0025), and The State of Hawaii Department of Land and Natural Resources (Endorsement No: I1393). Nathan Trump, General Manager, Island Harvest Inc., provided written permission to collect specimens on their property, and collections at the Pahala site were made under the auspices of a long-standing verbal permission historically granted by Randy Cabral and Randy Mochizuki, area managers, Mauna Loa Macadamia Nut Corp.

Three different methods were employed to collect beetles. The first method involved the use of *Ricinus communis* (castor bean) "trap" logs. *Ricinus communis* logs (diameter 7–15 cm) were cut to a length of 30–35 cm, and both cut ends were dipped into paraffin wax to reduce the drying out of the logs. A quercivorol lure (ChemTica International S.A., Costa Rica), a known attractant of the beetles (*Carrillo et al., 2015*; *Dodge et al., 2017*), was attached to a bundle of six logs (to attract beetles) and this bundle was then hung in the field. The logs were left for 8 weeks to allow ample opportunity for foundresses to locate them and initiate their galleries. The logs were then retrieved and placed in laboratory cages. Beetles were collected daily as they emerged from the galleries in the logs. These logs were deployed under *Leucaena* trees at the Waimānalo Research Station, and in a mature
**Table 1   Identity of Hawaiian specimens of the *Euwallacea fornicatus* species complex collected from Oʻahu and the Big Island using a variety of collection methods.** All individuals were first identified as *E. perbrevis* (TSHB) or *E. fornicatus* (PSHB) using a high-resolution melt assay (Fig. 1) and the diagnosis of a subset of these was confirmed by sequencing the COI gene. The number of specimens used for the COI sequencing are indicated between brackets.

| Location | Date | Collection method | TSHB | PSHB |
|---|---|---|---|---|
| **Oʻahu:** | | | | |
| Maunawili | 7/2/2019 | Querciverol baited Castor bean trap log | 1 | 9 |
| Waimānalo | 7/26/2019 | Quercivorol baited Castor bean trap log | 1 | 9 (2) |
| Mānoa | 3/13/2018 | Extracted from Monkeypod branches | 2 (2) | 1 |
| Waiʻanae Mnts—Honouliuli Forest Reserve | 3/18/2019 | Extracted from *Planchonella sandwicensis* branches | – | 3 (3) |
| Kahana | 10/1/2019 | Extracted from *Hibiscus* branches | 8 (3) | 2 |
| Koʻoalau Mnts, Poamoho Tr. | 3/18/2019 | Ethanol and methanol baited Lindgren funnel traps | 2 (2) | – |
| **Big Island:** | | | | |
| Island Harvest Inc., Kapaʻau | 10/31/2019 | Extracted from dead macadamia branches | 9 (2) | 1 |
| Hilo Forest Reserve (Laupāhoehoe Experimental Forest Unit) | 1/11/2019 | Ethanol and methanol baited Lindgren funnel traps | – | 2 (1) |
| Waiākea Research Station - USDA field 3 | 12/4/2019 | Quercivorol baited funnel traps in Macadamia orchard | 6 | – |
| Waiākea Research Station - UH field 1 | 12/4/2019 | Quercivorol baited funnel traps in Macadamia orchard | 4 | – |
| Waiākea Research Station - UH field 2 | 12/4/2019 | Quercivorol baited funnel traps in Macadamia orchard | 2 | 1 |
| Pahala, Kaʻū | 12/4/2019 | Quercivorol baited funnel traps in Macadamia orchard | 3 (3) | 79 (2) |

castor bean grove at Maunawili (Table 1). These sites were about 9.5 km apart on the eastern side of Oʻahu. Trapping took place from the beginning of June 2018 until the end of September 2019. Beetles were also captured using Lindgren funnel traps. In the Koʻolau Mountains (Oʻahu) and the Hilo Forest Reserve (Big Island), the traps were "baited" with approximately 150 mL of an ethanol–methanol solvent lure containing between 40–50% ethanol and 50–55% methanol (Klean-Strip® Denatured Alcohol, W.M. Barr & Co. Inc., Memphis, TN, USA), and 50 mL of commercial anti-freeze car coolant containing ethylene glycol (Table 1). At the Waiākea Research Station and around Pahala, in the Kaʻū district of the Big Island, Lindgren funnel traps baited with quercivorol lures were deployed in macadamia orchards (Table 1). Finally, in several areas, beetles were also extracted directly from infested branches of a variety of different host plants including macadamia, hau (*Hibiscus tiliaceus*), monkeypod (*Samanea saman*), and, in the Waiʻanae mountains of Oʻahu, the endemic *Planchonella sandwicensis* (Table 1).

## Identification of specimens

Two methods were used to determine the identity of the beetles. Initial identifications were made using a high-resolution melt (HRM) assay similar to that described by *Rugman-Jones & Stouthamer (2017)*. Consistent, species-specific differences have been reported between TSHB and PSHB (and KSHB) in the DNA sequences of the 28S ribosomal subunit (*Stouthamer et al., 2017*). Thus, PCR primers were designed for a short fragment of DNA spanning a particularly variable region of 28S (GenBank accessions MT822790, MT822791, MT822792; Fig. 1A). This resulted in the primer pair, *P-K-Tfor* (5′-CGATCTCTGGCGACTGTTG-3′) and *P-K-Trev* (5′-GGTCCTGAAAGTACCCAAAGC-3′), which yielded diagnostic melt curves for TSHB, PSHB, and KSHB (Fig. 1B). DNA was

**A)**

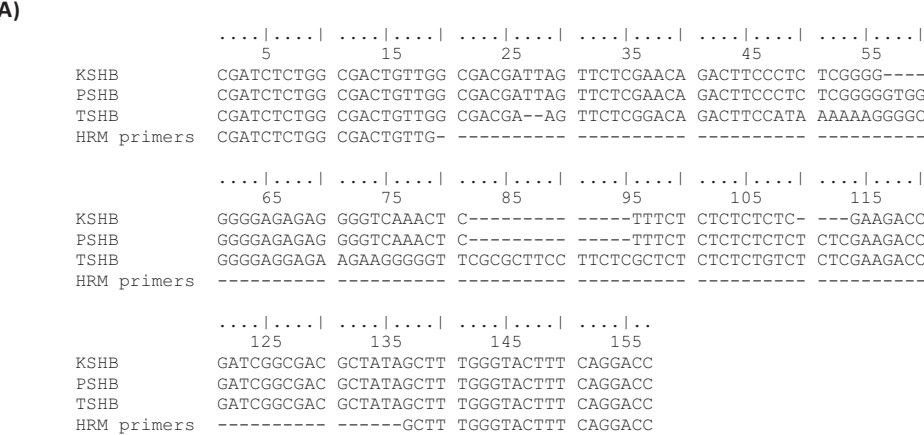

```
                ....|....| ....|....| ....|....| ....|....| ....|....| ....|....|
                     5         15        25        35        45        55
       KSHB     CGATCTCTGG CGACTGTTGG CGACGATTAG TTCTCGAACA GACTTCCCTC TCGGGG----
       PSHB     CGATCTCTGG CGACTGTTGG CGACGATTAG TTCTCGAACA GACTTCCCTC TCGGGGGTGG
       TSHB     CGATCTCTGG CGACTGTTGG CGACGA--AG TTCTCGGACA GACTTCCATA AAAAAGGGGC
       HRM primers CGATCTCTGG CGACTGTTG- ---------- ---------- ---------- ----------

                ....|....| ....|....| ....|....| ....|....| ....|....| ....|....|
                     65        75        85        95        105       115
       KSHB     GGGGAGAGAG GGGTCAAACT C--------- -----TTTCT CTCTCTCTC- ---GAAGACC
       PSHB     GGGGAGAGAG GGGTCAAACT C--------- -----TTTCT CTCTCTCTCT CTCGAAGACC
       TSHB     GGGGAGGAGA AGAAGGGGGT TCGCGCTTCC TTCTCGCTCT CTCTCTGTCT CTCGAAGACC
       HRM primers ---------- ---------- ---------- ---------- ---------- ----------

                ....|....| ....|....| ....|....| ....|..
                     125       135       145       155
       KSHB     GATCGGCGAC GCTATAGCTT TGGGTACTTT CAGGACC
       PSHB     GATCGGCGAC GCTATAGCTT TGGGTACTTT CAGGACC
       TSHB     GATCGGCGAC GCTATAGCTT TGGGTACTTT CAGGACC
       HRM primers ---------- ------GCTT TGGGTACTTT CAGGACC
```

**B)**

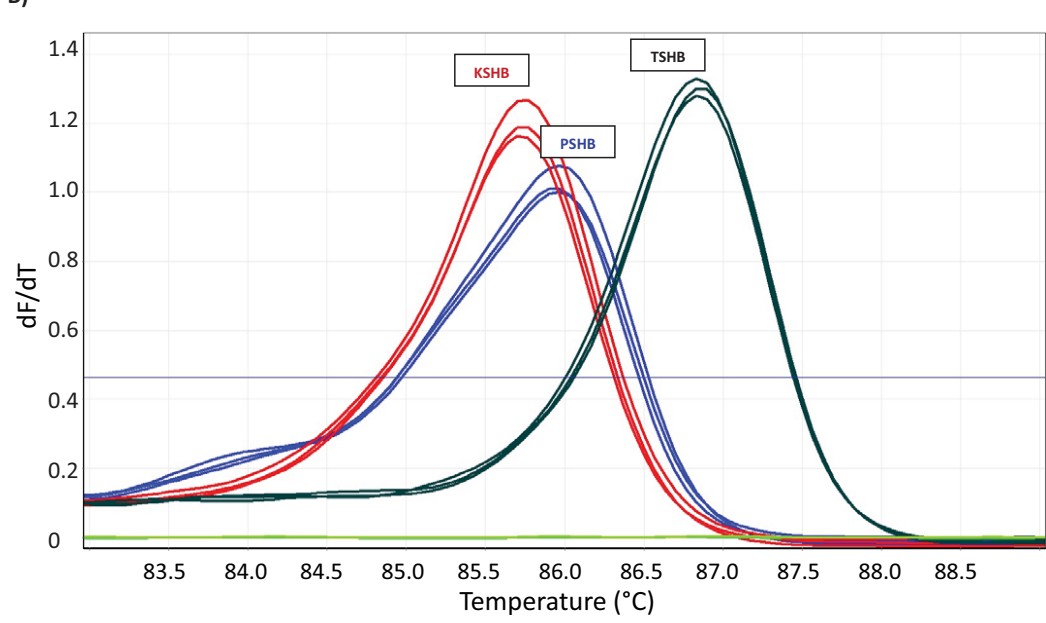

**Figure 1** **High-resolution melt (HRM) assay used to diagnose three species of the *Euwallacea fornicatus* species complex; *E. kuroshium* (KSHB), *E. fornicatus* (PSHB), and *E. perbrevis* (TSHB).** HRM was performed on a Rotor-Gene Q (QIAGEN), immediately after amplification of a species-specific fragment of the 28S ribosomal subunit (A) (GenBank accessions MT822790, MT822791, MT822792), and yielded characteristic melt curves (B). Three individuals are shown per species. Only TSHB and PSHB are known to occur in the state of Hawai'i.

extracted from individual beetles using the simple, non-destructive HotSHOT method (*Truett et al., 2000*), resulting in a final volume of 200 μL. The HRM utilized a Rotor-Gene Q 2-Plex qPCR machine (QIAGEN) and reactions were performed in 20 μL volumes containing 1× HOT FIREPol® EvaGreen® HRM Mix (Mango Biotechnology, Mountain View, CA, USA), 0.2 μM each primer, and 2 μL of DNA template. After an initial denaturing step of 95 °C for 15 min (required to activate the HOT FIREPol® DNA Polymerase),

amplification was achieved via 40 cycles of 95 °C for 20 s, 57 °C for 30 s and 72 °C for 30 s. Immediately following amplification, a melt analysis was conducted. PCR products were held at 77 °C for 90 s and then heated in 0.1 °C increments to a final temperature of 92 °C. Reactions were held for 2 s at each temperature increment before fluorescence was measured. Duplicate reactions were run for each specimen and positive controls for each of the three species were included in each run, as were 'no-template controls'. Based on the outcome of the HRM assays, the DNA of a subset of twenty specimens was sequenced to confirm its HRM diagnosis, and thereby validate the HRM assay. The COI gene was amplified from the HotSHOT-extracted DNA using the primers LCO1490 and HCO2198 'barcoding' primers (*Folmer et al., 1994*) following *Stouthamer et al. (2017)*. Purified amplicons were direct-sequenced in both directions at the Institute for Integrative Genome Biology, UCR.

In an attempt to identify the potential native origin of the Hawaiian PSHB population (see Results) its COI sequence (haplotype) was compared with those of native PSHB populations surveyed in previous studies (*Stouthamer et al., 2017*; *Gomez et al., 2018*; *Smith et al., 2019*). The respective sequences were retrieved from GenBank, combined with the Hawaiian sequences and collapsed into haplotypes using DnaSP version 5.10.01 (*Librado & Rozas, 2009*). The H8 haplotype of *E. perbrevis* (*Stouthamer et al., 2017*; *Smith et al., 2019*) was added to root the analysis, and the entire dataset was trimmed to 567bp. Genealogical relationships among the haplotypes were investigated by conducting a maximum likelihood (ML) analysis in RAxML version 8.2.10 (*Stamatakis, 2014*) using the RAXMLGUI v. 2.0.0.-beta6 (*Edler et al., 2019*). The program jModeltest 2.1.4 (*Darriba et al., 2012*) was used to identify GTR + $\Gamma$ + I as the best-fit model of nucleotide substitution. The dataset was partitioned by third codon position and node support was assessed with 1,000 rapid bootstrap replicates. The resulting tree was redrawn using FigTree v.1.4.3 (http://tree.bio.ed.ac.uk/software/figtree/).

## Identification of fungal species isolated from PSHB specimens

Using a method similar to that described by *Lynch et al. (2016)*, fungal species associated with PSHB were isolated from the heads of female beetles collected alive from funnel traps in macadamia orchards around Pahala, Big Island. Beetles were surface sterilized by submerging in 70% ethanol and vortexing for 20 s. They were then rinsed with sterile de-ionized water and allowed to dry on sterile filter paper. Individual beetles were decapitated under a dissection microscope, and the head (containing the mycangia) was macerated in a 1.5 mL microcentrifuge tube using a sterile plastic pestle. Each macerated head was suspended in 1 mL of sterile water and 25 μL of this suspension was pipetted onto a Petri plate containing potato dextrose agar (PDA; BD Difco, Sparks, MD) amended with 0.01% (w/v) tetracycline hydrochloride (PDA-t) and spread using sterile glass L-shaped rods. Plates were incubated for 48-72 h at 25 °C and single spore fungal colonies with unique morphologies were sub-cultured and shipped to the Eskalen lab (UC Davis) for molecular identification. The remaining abdomen/thorax segments were shipped to the Stouthamer lab (UC Riverside) for molecular identification of the beetle. DNA was extracted from the fungal isolates and sequenced following protocols detailed by *Carrillo et al. (2019)*,

in which the PCR primers ITS4 and ITS5 (*White et al., 1990*) were used to amplify the ITS1-5.8S-ITS2 region of the fungal ribosome. Beetles were extracted and sequenced as described above.

## RESULTS

A total of 145 beetles were examined in this study. Of these, the HRM assay identified 38 as TSHB and 107 as PSHB (Table 1). COI sequences of a subset of twenty of these specimens (12 ×TSHB and 8 ×PSHB) confirmed their HRM diagnosis, providing validation for the remaining 125 diagnoses. The COI sequence of the TSHB individuals was identical to that of all earlier TSHB specimens from Hawaiʻi, matching the H8 haplotype from *Stouthamer et al.* (*2017*; GenBank accession KU726996). Similarly, the PSHB sequences were also all identical to the sequence belatedly deposited in GenBank for specimens collected from macadamia on the Big Island in 2007 (*Mitchell & Maddox*, *2010*; GenBank accession KX818247). Both species were found on the two islands surveyed; the Big Island and Oʻahu (Table 1). PSHB appeared to be particularly abundant on the Big Island, accounting for 80 of the specimens trapped in Lindgren traps placed in macadamia orchards as opposed to only 15 TSHB. A further 9 TSHB and 1 PSHB were extracted from dead macadamia branches. Only two specimens were trapped in the Hilo Forest Reserve both of which were PSHB. The relative abundance of the two species was slightly more balanced in our Oʻahu surveys with a total of 24 PSHB and 14 TSHB (63% and 37%, respectively). The Hawaiian PSHB haplotype did not match any from the native range but in the ML analysis it grouped with haplotypes from Vietnam, Thailand, and China (Fig. 2).

Fungi were identified from the heads of four individual specimens from the Big Island. Sequences of the COI confirmed these specimens as PSHB and both *Fusarium euwallacea* and *Graphium euwallacea* were identified. Both fungi were successfully cultured from two of the specimens and of the remaining specimens, only *G. euwallacea* was successfully cultured from one, and only *F. euwallacea* was cultured from the other. The DNA sequences obtained for the ITS1-5.8S-ITS2 region of these fungi were identical (100%) to those obtained for the fungi associated with PSHB in California (GenBank accessions JQ723754 (*F. euwallaceae*) and KF540225 (*G. euwallaceae*)). In addition to these fungi, one specimen harbored a further *Fusarium* sp. previously associated with the decline of Indian coral tree, *Erythrina variegata*, on the island of Okinawa, Japan (GenBank accession LC198904).

## DISCUSSION

The islands of the US state of Hawaiʻi are particularly prone to invasion by exotic species due to their geography, climate, history, and economy. Indeed, it is thought that over half of Hawaiiʻs free-living species are non-indigenous (*US Congress, 1993*), and their numbers continue to rise. For example, a 1992 report documented the arrival of an average of 20 exotic invertebrate species each year from 1961 through 1991 (*The Nature Conservancy of Hawaiʻi, 1992*). Many of these species have had little noticeable effect in their new environment, but unfortunately a substantial proportion of adventive species have significantly impacted the ecology and economy of Hawaiʻi

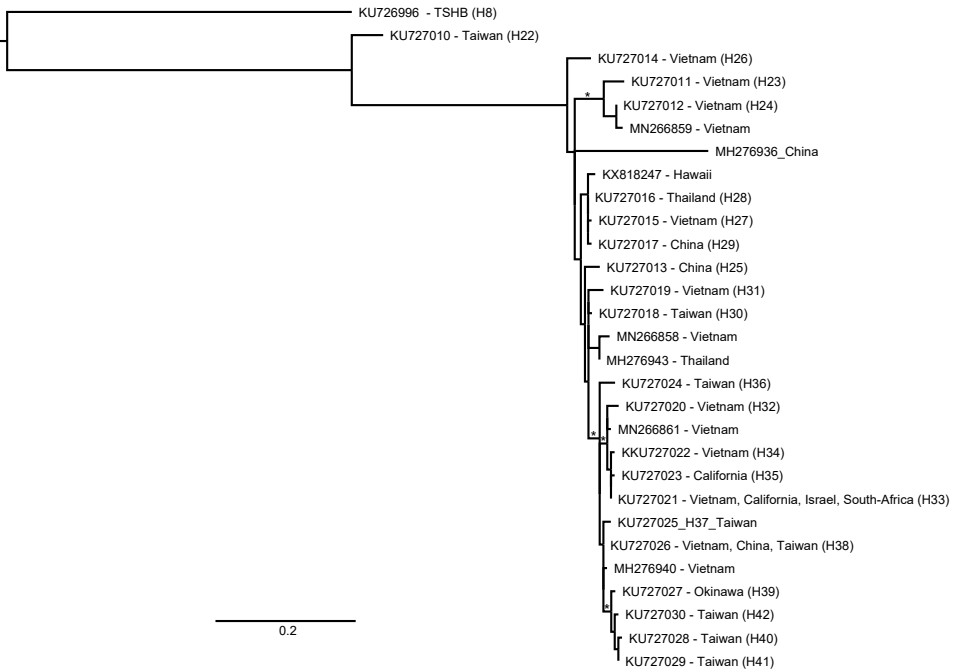

**Figure 2** **Relationship of an invasive Hawaiian population of *Euwallacea perbrevis* with those in its native range, based on COI sequences.** ML reconstruction performed in RAxML using all haplotypes deposited in GenBank (*Stouthamer et al., 2017*; *Gomez et al., 2018*; *Smith et al., 2019*). Branch support was assessed with 1,000 rapid bootstrap replicates. An asterisk denotes bootstrap support over 70%.

(*The Nature Conservancy of Hawai'i, 1992*; *State of Hawai'i, 2017*). The problems associated with detecting and accurately documenting invasive species are further complicated by the phenomenon of cryptic species: instances where genetically discrete species are erroneously classified as a single species because they are morphologically identical. This study provides the first confirmation that the Hawaiian Islands have been invaded not as previously thought by just one member of the *Euwallacea fornicatus* species complex, *E. perbrevis* (TSHB), but also by a second, *E. fornicatus* (PSHB).

Based on genetic characterization (HRM and sequencing) of beetles captured using a variety of different methods, it is clear that TSHB and PSHB occur on the Big Island and on Oʻahu. The co-occurrence of different cryptic species is not uncommon in this species complex (*Stouthamer et al., 2017*; *Gomez et al., 2018*; *Smith et al., 2019*). For example, in Taiwan, at least three species occur in complete sympatry (*Carrillo et al., 2019*). *Stouthamer et al. (2017)* previously confirmed the presence of TSHB on the Big Island and on Maui but during the process of publishing that study, a sequence was deposited in GenBank by another group of researchers, indicating that PSHB had been detected on the Big Island. Although the sequence was only deposited in September of 2016, it originated from two specimens captured in 2007 as part of a study investigating pests of macadamia (*Mitchell & Maddox, 2010*). Just how long PSHB, or indeed TSHB, have been present in Hawaiʻi remains unknown, but the current study confirms that both are well-established.

It would be interesting to examine the entomological collections of the Bishop Museum, Honolulu, to look for evidence of the arrival of both. Recent systematic studies (*Gomez et al., 2018*; *Smith et al., 2019*) show that TSHB and PSHB can be separated from each other based on certain morphometric measurements, so these collections may hold a historical record of these invasions. The authoritative public website http://www.barkbeetles.info/ identifies specimens collected on Oʻahu from *Erythrina* sp. in 1919 as *E. perbrevis* (TSHB), presumably based on morphometric measurements. Since our survey targeted only the two most-populous islands, it is unknown whether PSHB is also present on the other islands. The species complex has also been recorded on Kauaʻi, Maui, and Molokaʻi (*Swezey, 1941*; *Schedl, 1941*; *Samuelson, 1981*). As mentioned, TSHB is known from Maui (*Stouthamer et al., 2017*) but it now seems possible that PSHB is also there. As for the other two islands, no beetles have been sequenced from either Kauaʻi or Molokaʻi, so the specific identity of those members of the *E. fornicatus* species complex remains a mystery.

The PSHB haplotype identified in this study (identical to KX818247) has not been identified from the native area of the species complex (*Stouthamer et al., 2017*; *Gomez et al., 2018*; *Smith et al., 2019*). As such, it provides little information for identifying the area of origin of the Hawaiian invasion. The Hawaiian haplotype was most similar to the H27, H28, and H29 haplotypes of *Stouthamer et al. (2017)* which were identified from populations in Vietnam, northern Thailand, and China, respectively (Fig. 2). This more or less encompasses the entire native range of PSHB, as we currently understand it, excepting the islands of Taiwan, Okinawa, and Hong Kong.

DNA sequences of the symbiotic fungi recovered from the Hawaiian beetles also provided little information about potential origin. The *F. euwallaceae* and *G. euwallaceae* sequences generated from Hawaiian PSHB have, as yet, never been recovered in the native area of the beetles (*Carrillo et al., 2019*). However, the Hawaiian fungal sequences were identical to those of the fungi associated with the invasive PSHB populations in California (*Eskalen et al., 2012*; *Lynch et al., 2016*) and Israel (*Freeman et al., 2013*). This creates an interesting paradox. Invasive PSHB populations in California and Hawaiʻi likely have different origins within the beetle's native range, and yet share identical *Fusarium* and *Graphium* fungal symbionts, neither of which have been isolated from PSHB anywhere in its native range. Indeed, among invasive populations of the *E. fornicatus* species complex, only the *Fusarium* associated with KSHB in California, *F. kuroshium*, has been found in the native range in Taiwan, although to add further to the conundrum, in Taiwan it has only been isolated from PSHB, and not KSHB (*Carrillo et al., 2019*).

Whatever its origin, it currently appears that the Hawaiian haplotype of PSHB has only invaded the Hawaiian Islands, where economic or ecological damage have yet to be quantified. However, history suggests that we should perhaps not ignore its presence. PSHB was first detected in California as early as 2003 but was not recognized as a problem until 2012 (*Eskalen et al., 2013*). One of the two haplotypes identified in California (H33; *Stouthamer et al., 2017*) has also successfully established in Israel and South Africa, where, like in California it is significantly impacting both agriculture and native ecosystems. Exactly why beetles with this particular haplotype are such successful and widespread invaders is unclear. Perhaps it is just evidence of a serial invasion, with global trade aiding the

subsequent movement of one established invasive "bridgehead" population to other areas. But it may also be linked to differences in the virility of different haplotypes and/or the symbiotic fungi they carry. The fungal species carried by the Hawaiian beetles are identical to those carried by H33 which have proven pathogenic to a multitude of tree species (*Eskalen et al., 2013*). The impact on native Hawaiian vegetation is at this point minor, or unrecognized, but several notable endemic species are attacked including *Acacia koa*, *Pipturus albidus,* and *Planchonella sandwicensis* (Gillett *pers.obs.*). Other host plants known to be used by beetles belonging to the *E. fornicatus* complex in Hawai'i include, *Albizia lebbek*, *Albizia moluccana, Aleurites moluccana, Artocarpus altilis, Citrus, Colvillea, Cucumis, Enterolobium cyclocarpum, Eugenia jambolana, Ficus, Leucaena, Litchi chinensis, Macadamia, Mangifera, Nothopanax guilfoylei, Persea gratissima, Ricinus communis, Samanea, Schinus molle, Spondias, Sterculia foetida,* and *Tamarindus* (*Samuelson, 1981*).

While this study confirms that two members of the *E. fornicatus* species complex, TSHB and PSHB, have successfully established on the Hawaiian Islands, the full geographic extent of the two species remains unknown, since our survey focused only on the Big Island and O'ahu. Furthermore, we focused our efforts on particular crops (macadamia) and locales. In our captures, PSHB was more abundant than TSHB but this may not be an accurate reflection of the relative abundance of the two species across different habitats and islands. Detecting invasive species across a large and heterogeneous landscape presents difficult challenges and will require cooperation among many stakeholders. Without a monitoring program aimed specifically at the *E. fornicatus* species complex, relevant agencies might at least seek to collate any by-catch specimens from other programs, which match the morphological description of *E. fornicatus*. The diagnostic method developed herein, based on HRM, then provides an important tool allowing the quick, cheap, and accurate identification to species of three cryptic members of the *E. fornicatus* species complex. KSHB was not detected in the current sample, but may prove a good inclusion for any future survey work. This specific assay was based on a stretch of the 28S nuclear ribosomal gene that is typically well-conserved within a species. Unlike a previous assay that was based on the much more variable COI gene (*Rugman-Jones & Stouthamer, 2017*), and indeed was developed to identify such intra-specific variation, the new assay is unlikely to be affected by "unknown" intra-specific variation. Thus, it provides a more accurate means of species identification when "going in blind" (i.e., working in a new habitat without in-depth knowledge of intra-specific variation). Following identification via the HRM assay, subsequent sequencing of the COI of 20 of our specimens, confirmed their identity and validated the HRM assay.

## CONCLUSIONS

The present study has confirmed that two cryptic species from the *E. fornicatus* complex, TSHB and PSHB, have established invasive populations in the Hawaiian Islands. Both species are present on the Big Island and on O'ahu. Earlier work also confirmed the presence of TSHB on Maui, but documenting the full geographic extent of these invasive species in Hawai'i will require further survey work targeting the remaining islands, and

different habitats. Should researchers wish to pursue such research, we have developed a cheap and reliable molecular assay for accurate species diagnosis.

## ACKNOWLEDGEMENTS

CG and FY wish to thank Cynthia King, Ryan Peralta (both DLNR –DOFAW), and Tabetha Block (USDA) for their assistance in obtaining research permits. FY would also like to thank Tracie Matsumoto (USDA-ARS, Hilo, Waiākea Research Station, UH, Hilo) and Luiz (manager) Pahala macadamia farm, Kaʻū for allowing field trapping studies.

### Funding

Richard Stouthamer was supported by USDA-APHIS cooperative agreement AP19PPQS&T00C244. Mark G. Wright was supported by the State of Hawaiʻi Department of Agriculture (HDOA) and USDA-NIFA Hatch Project HAW09041-H. Conrad Gillett was supported by the State of Hawaiʻi Department of Agriculture (HDOA) and USDA State Research, Education and Extension project HAW00942-H. The funders had no role in study design, data collection and analysis, decision to publish, or preparation of the manuscript.

### Grant Disclosures

The following grant information was disclosed by the authors:
USDA-APHIS cooperative agreement: AP19PPQS&T00C244.
State of Hawaiʻi Department of Agriculture (HDOA).
USDA-NIFA Hatch Project: HAW09041-H.
State of Hawaiʻi Department of Agriculture (HDOA).
USDA State Research, Education and Extension project: HAW00942-H.

### Competing Interests

The authors declare there are no competing interests.

### Author Contributions

- Paul F. Rugman-Jones conceived and designed the experiments, analyzed the data, prepared figures and/or tables, authored or reviewed drafts of the paper, and approved the final draft.
- Michelle Au, David Honsberger and Deena Husein performed the experiments, authored or reviewed drafts of the paper, and approved the final draft.
- Valeh Ebrahimi performed the experiments, analyzed the data, authored or reviewed drafts of the paper, and approved the final draft.
- Akif Eskalen conceived and designed the experiments, performed the experiments, analyzed the data, authored or reviewed drafts of the paper, and approved the final draft.
- Conrad P.D.T. Gillett, Mark G. Wright and Fazila Yousuf conceived and designed the experiments, performed the experiments, authored or reviewed drafts of the paper, and approved the final draft.

- Richard Stouthamer conceived and designed the experiments, analyzed the data, authored or reviewed drafts of the paper, and approved the final draft.

## Field Study Permissions

The following information was supplied relating to field study approvals (i.e., approving body and any reference numbers):

Specimens were collected with the following permissions: Scientific research and collecting permit for Hawaii Volcanoes National Park (permit # HAVO-2019-SCI-0025), issued to Conrad Gillett by the United States Department of the Interior National Park Service; Natural Area Reserve and native invertebrate research permit (Endorsement No: I1393), issued to Conrad Gillett by The State of Hawaii Department of Land and Natural Resources; Nathan Trump, General Manager, Island Harvest Inc., provided written permission to collect specimens on their property. Collections at the Pahala site were made under the auspices of a long-standing verbal permission granted by Randy Cabral and Randy Mochizuki, area managers, Mauna Loa Macadamia Nut Corp.

## DNA Deposition

The following information was supplied regarding the deposition of DNA sequences:

The COI sequences of beetles reported in this study match GenBank KU726996 (TSHB) and KX818247 (PSHB). The fungal sequences match JQ723754 (*F. euwallaceae*) and KF540225 (*G. euwallaceae*).

PCR primers for the HRM assay were designed using 28S sequences available at GenBank: MT822790, MT822791, MT822792.

## Data Availability

Count data is available in Table 1.

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
