# Peer review of "One becomes two: second species of the Euwallacea fornicatus (Coleoptera: Curculionidae: Scolytinae) species complex is established on two Hawaiian Islands"

_PeerJ, doi:10.7717/peerj.9987_

## Round 0.1 · original submission · Minor Revisions

Two reviewers suggested minor revision, but one required major revisions. Please, provide a revised version considering the three reviews, but especially considering concerns by reviewer 3.

Reviewer 1 ·

Basic reporting

The manuscript is well-written and the approach is logical, It provides a thorough and well-written account of an overlooked discovery and subsequent confirmation of two species, rather than one as previously thought, of the Euwallacea fornicatus complex in Hawaii. The invasion of this species complex to several regions and its taxonomic dynamism makes the ms topical and relevant both in its report of the range expansion of one of the invasive species, and in its development of a method to easily differentiate taxa without sequencing. I enjoyed reading it very much and it adds important additional data and support to this fascinating system.

Experimental design

While not comprising an 'experiment' in the traditional sense, the approach is well-considered and thorough and the aims are clear. I am unfamiliar with the high-resolution melt approach, but it appears to provide a fast, accurate and repeatable method for differentiating species. All other aspects - trapping, collection, sequencing of beetles, isolation and sequencing of fungi - are well described and conducted.

Validity of the findings

The second species was independently collected, and was verified through additional collections, and sequencing, here. The HRM approach was validated using traditional sequencing and I am satisfied that both it, and the results reported here are correctly interpreted - I am in no doubt of their veracity.

Additional comments

I have only a few minor corrections to the ms here:
L3: delete “both” as it sounds like you only sampled two macadamia orchards
L25,L27: it doesn’t bother me because I am familiar with the acronyms, but it is unusual to present them without explaining what they stand for, particularly out of context as here. This is up to the editor, as I actually think it is simpler not to have to write in full “tea shot hole borer” and “polyphagous shot hole borer” – but it does contravene normal first-use of acronyms.
L32: delete “just”
L33: newly-confirmed
L35-36: delete “It is of particular note that” – being in the abstract makes it already noteworthy.
L38: delete “anywhere”
L42: delete “have”, “much recent” and “both” from the first sentence. It otherwise reads a little too America-centric since they are also invasive elsewhere and this has attracted recent attention too.
L57: why just Hawaii here? Maybe change to “these three states” or something as it seems odd to leave California and Florida out of it now.
L58: delete “of”
L86-87: capitalise Shot Hole Borer for PSHB and KSHB as for TSHB (or de-capitalise for TSHB)
L95: TSHB is not perbrevis in Gomez et al as cited here.
L117: delete “also”
L118: eastern
L121 and others: mL rather than ml
L192,193,194: is the species name euwallceae rather than euwallcea?
L221: delete “had”
L225: delete “actually”
L227: or indeed how long either species has been present. Is it not possible that the original records prior to 1910 could be PSHB?
L240: delete “subsequently” and “additional”
L245: is this better expressed by replacing “With the exception of” by “Along with”?
L250: replace “,as yet, never” with “not”
L261: delete “currently”
L305,306: move “TSHB and PSHB” from end of sentence to after “species”

Thank you for the opportunity to review the ms.

·

Basic reporting

A nicely written paper.

Experimental design

no comment. The diagnostic tool developed will help identify these species.

Validity of the findings

no comment.

Additional comments

1. Line 228-229. It is very unfortunate that no attempt was made to locate additional specimens of these species in either the Bishop Museum or in the UH Manoa collection (where one author on this manuscript is based). I understand that the Bishop may not be accessible at this point due to covid, but the UH collection should be easy for at least one author to check. The big Island and Oahu were surveyed as part of the Forest Service’s EDRR program back in 2009 (Rabaglia et al. 2019: https://academic.oup.com/ae/article/65/1/29/5376569; no need to cite this unless you find it necessary). The UH collection contains authoritatively identified specimens of both species from the 2009 which will help elucidate how widespread E. fornicatus was distributed in the Big Island around the time the Maddox specimen was sequenced. I identified these specimens in January 2020.

2. Line 230. These species are actually relatively easy to separate, especially when they are next to each other in a mixed series. See both Gomez et al. 2018 and Smith et al. 2019’s tables and note that the two species do not overlap in either size or length/width ratio. This is a fact and not a claim as it has been tested and these differences are statistically significant (see Gomez et al.). Rewrite.

3. The authors improperly refer to Euwallacea fornicatus with usage of s.s. and s.l. throughout the manuscript. The four species in the complex are monophyletic lineages and associated with species names. They should be referred to as these names or 'species complex' instead. It is acceptable in the intro but not in the abstract, lines 27, 64, 80, 98-99, 102, 216, Figure 1, or Table 1.

4. Line 305: The word ‘cryptic’ is not really applicable to these two species as they can easily be separated with a micrometer.

5. Line 51. Correctly: Réunion.

Reviewer 3 ·

Basic reporting

Minor rewrite necessary.
An additional figure is suggested.

Experimental design

Measuring additional specimens is suggested.
An additional analysis is suggested.

Validity of the findings

The findings are valid although the discussion does not consider the entirety of recently published studies and data.

Additional comments

Line 99 – “(according to Smith et al., 2019)” This is an odd phrase for a citation. Do the authors doubt the conclusions of this publication? If so more explanation is needed. If not, delete “according to”.

Line 216 – no need to include s.s. after E. fornicatus at this point in the paper. The use sensu stricto is reserved for when species boundaries are obscure. Gomez et al. 2018 and Smith et al. 2019 delimit the species (I read both papers so to be certain).

Line 218 – “ both are found..” simplify to occur.

Lines 227 -229 – Why not check the Bishop and the UH collections for specimens of PSHB ? One author is located in Oahu – I don’t think it is unreasonable (even with COVID -19 precautions) to request a loan of specimens from Bishop and the UH collections.

Lines 229- 232 – This sentence is awkwardly written and its meaning is unclear. 1. “Recent systematic studies (Gomez et al., 2018; Smith et al., 2019) have claimed that TSHB and PSHB…” What do the authors mean by “claimed”? This implies that the results/conclusions of the above two studies are suspect. If so this point would need some explanation. (Note: Upon reading these papers, I found the authors’ methodologies sound and conclusions reasonable. Their results clearly shows that the body sizes of these two species are distinct.) The authors do not use the morphometric analysis in this paper so the assertion that the conclusions of Gomez et al., 2018 and Smith et al., 2019 is baseless. 2. “… so these collections may have inadvertently captured a historical record of this invasion.” “Inadvertently” suggests that these entomology collections were made without purpose or haphazardly which is dismissive of their importance. I suggest this wording “…so these collections may hold a historical record of this invasion.”

Concerning my comments pertaining to Lines 227-232, I believe conducting morphometric analysis on suspect specimens of PSHB in the Bishop and UH collections would be greatly improved this paper it is an opportunity to discover the earliest record of this species in Hawai’i.

Lines 230 -242– I understand the authors point that the exact PSHB haplotype (identical to KX818247) has not been found in the native range. However, sequences similar to it may give some evidence to its origin. The authors suggest that KX818247 is close to H27-29 but there are more E. fornicatus sequences reported in Smith et al. 2019. Conducting a phylogenetic analysis of all available E. fornicatus sequences would better illustrate the relationship of KX818247 to the haplotypes discover in the native range of the species.

Review the literature cited – there are several typographical errors throughout.

---

## Round 0.2 · accepted · Accept

I find that the revision provided by the authors as acceptable as it is.